# Reproducibility of the Methods in Medical Imaging with Deep Learning.

**Attila Simkó**[1]                                                     ATTILA.SIMKO@UMU.SE
**Anders Garpebring**[1]
**Joakim Jonsson**[1]
**Tufve Nyholm**[1]
**Tommy Löfstedt**[2]                                                 TOMMY.LOFSTEDT@UMU.SE

[1] *Department of Radiation Sciences, Umeå University, Umeå, Sweden*
[2] *Department of Computing Science, Umeå University, Umeå, Sweden*

## Abstract

Concerns about the reproducibility of deep learning research are more prominent than ever, with no clear solution in sight. The Medical Imaging with Deep Learning (MIDL) conference has made advancements in employing empirical rigor with regards to reproducibility by advocating open access, and recently also recommending authors to make their code public—both aspects being adopted by the majority of the conference submissions.

We have evaluated all accepted full paper submissions to MIDL between 2018 and 2022 using established, but adjusted guidelines addressing the reproducibility and quality of the public repositories.

The evaluations show that publishing repositories and using public datasets are becoming more popular, which helps traceability, but the quality of the repositories shows room for improvement in every aspect. Merely 22% of all submissions contain a repository that was deemed repeatable using our evaluations.

From the commonly encountered issues during the evaluations, we propose a set of guidelines for machine learning-related research for medical imaging applications, adjusted specifically for future submissions to MIDL. We presented our results to future MIDL authors who were eager to continue an open discussion on the topic of code reproducibility.

**Keywords:** Reproducibility, Reproducibility of the Methods, Deep Learning, Medical Imaging, Open Science, Transparent Research

## 1. Introduction

Concerns about reproducibility are present in all research fields, however the rapid speed of advancement in deep learning makes it an especially sensitive area. Much of current machine learning research is exploratory, gathering new observations and expanding the scope of our knowledge on earlier, but often still relatively recent research that perhaps was not evaluated appropriately or extensively enough. This leads to a risk that important research remains unnoticed if it does not follow the current trends, or even worse it risks entire lines of research collapsing if one of their building blocks is later disproven (Sculley et al., 2018). A grim example is a study (Melis et al., 2018) which showed that many novel model proposals in NLP that claim to be *state-of-the-art* are outperformed by a simple LSTM model, if it is well-tuned.

These concerns need to be acknowledged, discussed, and overcome. Implementing empirical rigor is important, such as advocating for Open Science, and it is certainly a way forward for improved reproducibility. However, there are some aspects that can be addressed to speed up the progress (Bohr and Memarzadeh, 2020).

As one form of reproducibility (Claerbout and Karrenbach, 1992), the main body of the paper should aim for *reproducibility of the conclusions* (Bouthillier et al., 2019). But since so many details are needed in a text for a method to be reproducible (Renard et al., 2020), it might not be feasible to include everything, especially when there are page constraint. Despite the authors being well intentioned, there might be a lack of awareness on how to achieve reproducibility of conclusions (Bouthillier et al., 2019), and the commonly used solutions might achieve something different.

*Reproducibility of the method* or traceability means that on the exact same hardware, using the same dataset and random seeds, one should be able to generate the same results as were presented in the paper. Practically, this is what code sharing achieves (if done properly), and a common method used by researchers that pursue transparency. However it does not necessarily improve the reproducibility of the conclusions. First of all, submitting a repository is not a trade-off for focusing less on the main body of the paper (Raff, 2019), and many new concerns arise by making code publicly available, especially so when handling sensitive medical data. We believe that a well-managed, high-quality repository can improve the overall readability, reproducibility, and therefore also the impact of a submission; and similarly, a systematic low-quality repository can weaken the general submission. Conference organizers could take the initiative here, and set up a common direction for the quality of supplementary code repositories, to help those authors that aim for transparency and reproducibility.

We are by no means the first to raise concerns about the lack of empirical rigor in the field, see *e.g.* (Sculley et al., 2018; Jacobs and Bram van, 2019). Some empirical rigor has been proposed for reporting findings (Mongan et al., 2020), by Nature[1] and some large machine learning conferences (NeurIPS[2] (Pineda et al., 2020), MICCAI (Balsiger et al., 2021)) have recently introduced guidelines for code submissions. There is also a paper evaluating the checklist provided by NeuroIPS (Pineau et al., 2021) and other initiatives that shed light on unrepeatable research[3]. But this doesn't necessarily translate directly to other conferences, that are expected to have their own individual concerns, such as the sensitivity of the data for example. The proposed guidelines often pose requirements to the authors, and we have decided to focus only on suggestions, using the checklist[4] implemented by NeurIPS as a baseline, adjusting it to better fit MIDL with regards to commonly encountered sensitive data, and a goal for transparency. We believe that the evaluations we have made will add to the conversation, and that our proposals can help to address reproducibility at the conference level.

We believe that the conference Medical Imaging with Deep Learning (MIDL) is in a perfect position to implement and follow their own set of guidelines to help authors and help to improve the quality and reproducibility of research in the field.

---

1. https://www.nature.com/documents/GuidelinesCodePublication.pdf

2. https://nips.cc/Conferences/2021/PaperInformation/CodeSubmissionPolicy

3. https://www.paperswithoutcode.com/

4. https://github.com/paperswithcode/releasing-research-code

## 2. Why MIDL?

MIDL has always been an advocate of open research, with the author guidelines recommending open access and using an open review system since the first conference in 2018. This innate focus on transparency adds to the impact and the quality of the conference, and given the field the approach is understandable, as both medical imaging (MI) and deep learning (DL) are particularly sensitive to the reproducibility problem (Renard et al., 2020).

The MIDL organizers show an active interest regarding the issues of reproducibility, but there is currently little or no support for further evaluation of these supplementary material, making them vulnerable to poor quality, which affects the impact of the entire submission.

The open review system makes it easy to add evaluation metrics regarding the repositories for the review phase, and the system also allows for easy follow-up evaluations. This way, we can recognize more relevant concerns and the guidelines can be adjusted accordingly to better fit MIDL, and to better reflect the current state of machine learning research.

As the MIDL conference continues to grow, we think it is in a perfect position to shine more light and take a stand regarding the issues of reproducibility.

## 3. Materials and Methods

We evaluated the submissions to MIDL using a combination of already existing guidelines (NeurIPS and MICCAI, as mentioned in Sec. 1), and also recorded any other issues that we observed. From these results we propose a set of guidelines with primary focus on future MIDL submissions, but which would likely be useful for other conferences as well. The evaluations focus on data and code availability with an in-depth analysis of the corresponding code repositories, when available.

For 23 papers, some evaluations were not applicable (*e.g.* providing trained models if the paper focuses on loss functions), these papers were excluded from further evaluations.

### 3.1. Public content

We report if the paper has an official public repository and if it used a publicly available dataset for either training or evaluation. For code sharing, the most common option is a public Git repository, for instance on Github. If storing large trained models is an issue, Zenodo, AWS, OneDrive, Google Drive, Dropbox, huggingface, *etc.* are popular choices. The increasing number and variety of publicly available datasets would allow authors to evaluate their methods on public data even if they were trained on privately acquired datasets. This would increase transparency and allow future methods for easier comparison to the ones presented in the paper. The popularity of public datasets is boosted by public challenges, that maintain high quality datasets and often offer the top teams a cash prize and the opportunity to publish their findings in high impact journals[5].

---

5. https://grand-challenge.org/

### 3.2. Repository evaluation

Each available repository is graded using a 6-point system (find the full details in Appendix A). Based on a more general guideline for building repositories[6], the evaluation consist of the following metrics:

**Dependencies**   A list of all the packages that were used when training the model, including their *exact* version numbers. The most commonly used `pip` and `conda` both support exporting the packages to text files, while tools also exist to export only the necessary packages[7].

**Code for model training**   Code that builds the model architecture reported in the submission, using the reported loss functions, and given a path to the training data, it begins training using the provided data loaders (experiment log files are also accepted).

**Code for evaluation**   Code to evaluate the trained model using the metrics reported in the paper. Alternatively, simple examples performing predictions with the trained model.

**Trained model**   Access to the *exact* trained model or model's weights used for the evaluations reported in the paper. Access to at least one model highlighted in the conclusions if multiple were presented and evaluated.

**Documentation**   A document (such as, *e.g.*, a readme file on Github) describing what's available in the repository and how to use it. The documentation should detail the loading and pre-processing of the data, the required dependencies, training, evaluation, and where to find the trained models.

**Licensing**   A description on how the files from the repository can be used. Github supports selecting from commonly used licenses (such as MIT[8], Apache-2.0[9], GPL 3.0[10]). For further information about licenses we refer the readers to a related website[11]. Authors asking others to cite their papers if their code was used, was not considered as sufficient licensing.

### 3.3. Model repeatability

To easily train a model for the same task as described in a given submission, it is required to reference a public dataset that can be used for training, to list the correct dependencies, and to provide code to build and train the model. If these requirements are fulfilled, the model is deemed repeatable. This is only one form of reproducibility, showing only that a similar model performing a similar task can be trained by publicly available content. We used this metric as a proxy for reproducibility of the methods.

---

6. https://github.com/paperswithcode/releasing-research-code

7. https://github.com/drivendata/cookiecutter-data-science

8. https://choosealicense.com/licenses/mit/

9. https://choosealicense.com/licenses/apache-2.0/

10. https://choosealicense.com/licenses/gpl-3.0/

11. https://choosealicense.com/

Table 1: Results of the evaluations. The results "Has Repository" and "Public Data" evaluate all valid submissions, while the other metrics evaluate all available repositories. The standard error is also shown for the average score.

|  | 2018 | 2019 | 2020 | 2021 | 2022 |
|---|---|---|---|---|---|
| Number of submissions | 47 | 47 | 65 | 59 | 98 |
| Has Repository (%) | 29,8 | 29,8 | 43,1 | 57,6 | 74,5 |
| Public Data (%) | 57,4 | 57,4 | 80,0 | 69,5 | 74,5 |
| Dependencies (%) | 50,0 | 28,6 | 53,6 | 52,9 | 56,2 |
| Training Code (%) | 78,6 | 78,6 | 75,0 | 76,5 | 79,5 |
| Evaluation/Demo (%) | 71,4 | 78,6 | 82,1 | 82,4 | 79,5 |
| Trained model (%) | 28,6 | 21,4 | 28,6 | 29,4 | 21,9 |
| Documentation (%) | 78,6 | 57,1 | 67,9 | 67,6 | 68,5 |
| Licensing (%) | 71,4 | 35,7 | 57,1 | 38,2 | 50,7 |
| Model repeatability (%) | 35,7 | 21,4 | 42,9 | 50,0 | 43,8 |
| Average score | 4,00±1,81 | 3,00±1,52 | 3,73±1,58 | 3,58±1,74 | 3,64±1,55 |

## 4. Evaluations

We evaluated all full paper submissions to the MIDL conference between 2018 and 2022 using the proposed guidelines for evaluating repository reproducibility. The guidelines are included in Appendix A.

We evaluated if the data used was available to the public or not. Then the 6-point checklist was filled out, based on the quality of the repository with respect to if they addressed package dependencies, had training code available, had evaluation code available, if the trained models evaluated in the paper were available, and if documentation and licensing existed for the repository. The model repeatability was evaluated using the collected factors, namely: The availability of the data, the dependencies, and the training code.

The evaluation was performed in June 2022 and repeated in October 2022 as some repositories have been updated following the conference. The licensing information was collected afterwards, in February 2023.

## 5. Results

Between 2018 and 2022, there were a total of 316 submissions to MIDL. In Table 5 the percentage of available code repositories and using public datasets are reported for all submissions. While the six quality metrics and their average score is reported for all repositories. Our checklist in full detail can be found in the appendix A).

For transparency, we have decided to publish the individual results as well. As a disclaimer, our goal was never to criticize individual submissions, but to show year-by-year trends. However if you don't agree with our evaluations for a specific submission, feel free

to contact us for a revision of the score. The presented results are based on the summary in an online spreadsheet[12] collected until May 8, 2023.

## 6. Discussion

For reproducible research in machine learning a plethora of hyperparameters need to be detailed. To circumvent in-depth details in the main body of the paper, the code used for the research is often made publicly available. Our evaluations of the submissions for the MIDL conference show that this practice is getting increasingly more popular, however the published repositories show serious flaws (seen in Table 5) — hindering their practicality and effectiveness. Publishing code is encouraged by the conference organizers (and it is common for other conferences as well), but the repositories are not reviewed. Poor quality or completely empty repositories are not discouraged and well-documented repositories are not awarded, hence focusing on their quality is not important. We argue that for this reason, they show no signs of improvements over the years. In contrast, areas where guidelines are in place (links for code and data) show an improving trend year by year.

The frequency of using public datasets peaked in 2021. A possible reason for this could be the impact of the Covid-19 pandemic, which could have led many to seek public datasets during 2020 for their 2021 submissions, but to draw such a conclusion would require other data than what we have collected here.

A total of 64 repositories were deemed repeatable according to our requirements, of the 163 repositories (42%), merely 22% of all submissions.

A total of 13 repositories were found completely empty or with broken links. Possibly with the fear of being rejected, or due to privacy concerns, access to the repositories at the time of submission is often replaced by texts along the lines of "the implementation will be made publicly available at the time of publication". However, it appears that often such sentences remain empty promises, and also with no chance for the reviewers to follow up on them at the time of publication. Although the long-term maintenance can not be overseen by a conference, a quick evaluation of the repositories could be encouraged for reviewers. For instance, to see if the repositories exist at the time of review or at least by the camera-ready deadline. Following conference-specific requirements for designing a repository can lead to extra work in case the submission is rejected and has to be submitted elsewhere, however we believe following a general set of guidelines makes the repository easy to translate between submissions.

We have noticed that public datasets are often used without the proper use of citations, and without links. This makes it hard to understand if the dataset is public or not, without prior knowledge of the particular datasets. We do not mean to advocate against using privately collected datasets, but for such cases we propose evaluating on public datasets or at least mentioning alternatives that are comparable to the privately collected data used and are available for the interested reader. This helps the reproducibility of the results and conclusions, and still makes it possible to publish the code.

When training a model on privately acquired data, we warn against privacy violation attacks (Tariq et al., 2020) where attackers could get information about the training data through the trained model weights.

---

12. https://docs.google.com/spreadsheets/d/1ndMJORbcsByOfr6Cygo8-wxYmzWZFkzI2T34IH1EZW8

One reason for authors to hesitate could be the fear of having to maintain a published repository, which is a valid point. But a well-designed repository requires no maintenance after submission, unless the authors wish to adjust it long-term.

The submissions were originally evaluated approximately a month before MIDL 2022. A month after the conference we have revisited the empty repositories from that year and found that 4 out of 9 have been modified. These changes did not affect our conclusions, however the later evaluations have been presented in this work. A great opportunity of online repositories is the possibility to modify, expand and improve, however it should be suggested to the authors that the repository is a part of their submission, and should be focused on before the conference. Since the repositories are adjustable, the results of our presented work might also change over time, if authors decide to revisit their old repositories. Therefore we wish to keep both our results and our proposed guidelines similarly dynamic, revisiting the online spreadsheet regularly.

The set of values represented by MIDL, such as advocating open review, using public datasets and publishing repositories, have made it possible for us to perform these evaluations, therefore we believe that our conclusions and suggestions can also help MIDL in improving reproducible research.

## 7. Conclusions

We have evaluated all submissions to the MIDL conference since its inception in 2018, and through 2022, and propose a reproducibility checklist for machine learning researchers with a focus on medical imaging. The checklist can be found in Appendix A.

To help conference organizers and attendees, we propose that the review process should include a quick evaluation of published repositories as well, just so that submissions with empty repositories and broken links can be addressed. We propose additional optional fields in the OpenReview submission form that follow the proposed checklist, addressing the availability and the reproducibility of the research. Reviewers should address when publicly available data is reported as future work, and point out that it adds no meaningful contribution to the research without it actually being available. The European Conference on Machine Learning and Data Mining (ECML-PKDD) allows authors to flag their submission as "reproducible", which requires posting evaluated code and the data, but it is rewarded by the conference in return. We suggest that MIDL acknowledges submissions that fill out the checklist and repositories that fulfill all the requirements.

To help authors, we propose to follow a reproducibility checklist, *e.g.* the one in Appendix A, when preparing a submission and a corresponding repository. Despite the possibility to address the code and data availability in the OpenReview process, we highlight the importance of addressing these in the main body of the paper as well.

We hope that with adequate guidance, such as a reproducibility checklist, many aspects of the reproducibility concerns surrounding deep learning methods can be resolved already at the submission or latest at the peer-review level. We further hope that this sparks a conversation about other aspects of reproducibility that also needs to be addressed, which would be very beneficial for the research field as a whole.

## 8. Preliminary impact

After contacting the MIDL board they suggested us to present our results to a wider audience within the framework of the MIDL Autumn Academy as an interactive session. This provided a great opportunity to get extremely valuable feedback from authors in the field. A poll regarding their experiences with code repositories was filled out by 81 of the participants (from an average total of 90 participants).

For building their own code repositories, 98% of the participants believe that the quality of a supplementing code repository affects the impact of the overall submission. Despite the perceived importance of the code quality, only 28% of the participants received help or feedback from a colleague or supervisor, while the others had to figure out how to build their repositories themselves, and only 53% said they had a good understanding on what content to include and how. 82% of the participants who had previously built a repository never got any feedback on it, neither from colleagues nor from other researchers. This lack of feedback is essential to address if we want the code quality to improve.

About using the code repositories of other researchers, only 4% of the participants (3 people) had never encountered any of the issues covered by the guidelines, which further underlines the importance of our work. However, 58% of the participants claimed to have encountered issues outside of the proposed guidelines. These include more practical issues (*e.g.*, hardware differences, using a mixture of programming languages, and bugs in the code) which clearly shows areas where the proposed guidelines could be further improved.

To motivate the rationality of our proposed checklist, we have collected the number of citations of each paper with a code repository—from Google Scholar, serving as a metric for the impact of the paper—excluding submissions from 2022 (the submissions are relatively new and therefore have a small number of citations). The average number of citations is plotted against our proposed reproducibility in the appendix in Fig. 1. Completely empty repositories (that received a score of 0) have the lowest, while repositories with a score above 3 generally have a larger average number of citations, further showing a connection between the quality of the code and the impact of the paper.

Additionally, during the discussions the participants agreed that integrating a reproducibility guideline into MIDL would be of great benefit to both the authors and for the impact of the manuscript; and further, that such an integration should not manifest as punishing incomplete repositories, but to reward reproducible code.

## 9. Future work

A logical next step would be to train and evaluate the models that are deemed repeatable, to see if we face any issues during the implementation of the available code. Through further discussions, the guidelines should be adjusted to cover commonly encountered practical issues that are not included yet in the checklist (*e.g.*, bugs in the code).

As reproducibility is becoming a more and more important part of future MIDL editions, we wish to further adjust these guidelines to help authors improve the impact of their submissions through high-quality code.

## 10. Acknowledgements

We are grateful for the financial support obtained from the Cancer Research Foundation in Northern Sweden (LP 18-2182, LP 22-2319, AMP 18-912, AMP 20-1014), the Västerbotten regional county, and from Karin and Krister Olsson.

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

## Appendix A. Proposed Reproducibility Checklist

Authors could benefit from answering the following questions about their submitted papers, to help readers interested in reproducing the presented results.

☐ **Is the reproducibility of the work addressed?** [Yes, No]
Detail in the main body of the paper the publicly available materials from the submission. Is there published code with the submission to make reproducing the results easier? Are there publicly available datasets to train a similar model? Is the trained model public?

☐ **Is the code publicly available?** [Yes, No]
Do property rights allow the authors to make their code publicly available? If so, include a link to the repository that stores the code used for the project. Details about how to access the code should be contained in the main body of the paper, and not only in the paper submission form. The repository should be available long-term.

☐ **Are public datasets used?** [Yes, No]
Address here, if the dataset was collected for the project and is not made publicly available. If the training dataset is private, aim to evaluate on public datasets for comparability. Alternatively, mention if there are similar publicly available datasets for reference.

☐ **Repository: Are the required package dependencies listed?** [Yes/No/NA]
The packages that have been used to achieve the reported results, and their version numbers. Without exact version numbers, the repositories become more difficult to use and therefore lose their value over time.

☐ **Repository: Is the code for model training available?** [Yes/No/NA]
Code for building the model with the exact same hyper-parameters and loss functions as reported in the paper, together with the training method.

☐ **Repository: Is the code for model evaluation available?** [Yes/No/NA]
Code for evaluating the trained model with the metrics presented in the paper.

☐ **Repository: Is the presented trained model available?** [Yes/No/NA]
If the trained model can be made publicly available, include the trained weights in the repository. Common formats are .pt for PyTorch, .h5 for TensorFlow, .pb for TensorFlow frozen graphs, or .onnx as an open format.

☐ **Repository: Is there documentation for the available material?** [Yes/No/NA]
A thorough and detailed description of the repository can be a major help for the interested reader to fully understand the code. Aim to describe the methods in a similar way as in the main body of the paper for coherence.

☐ **Repository: Is there licensing for the available material?** [Yes/No/NA]
A detailed description on how the shared material can be used for research and commercial purposes.

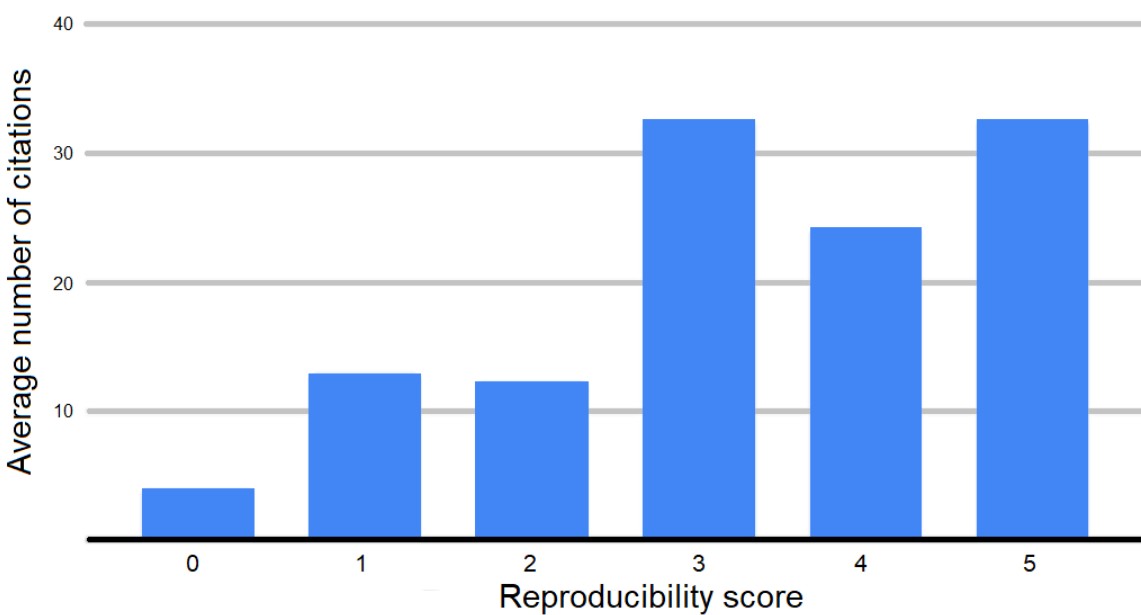

Figure 1: Results for evaluating the impact of the papers plotted against our proposed reproducibility score. The average number of citations were collected from Google Scholar for submissions with code repositories published before 2022.

