# OpenReview forum: "Reproducibility of the Methods in Medical Imaging with Deep Learning."
_MIDL.io/2023/Conference — MIDL 2023 Oral_

### Official Review · Reviewer_k5RB · 2023-02-04

**Confidence:** 4
**Preliminary Rating:** 5
**Recommendation:** Oral

**Summary:**

This paper presents an analysis of the reproducibility of all so far 316 accepted MIDL papers from 2018 and 2022. The evaluation showed that only 22% of the submission contain a repository that deemed repeatable according to their evaluation. Furthermore, the authors present a reproducibility checklist that could be implemented for future MIDL conferences.

**Strengths:**

- The paper is well-written and explains in very detail why reproducibility is important.
- The authors performed an extensive evaluation of 316 papers submitted to MIDL between 2018 and 2022 and checked all for repositories and evaluated them for five criteria: dependencies, code for model training, code for evaluation, trained model, documentation
- The results are stored in Google Sheets where everyone could check for each paper which score was reached. The authors promise to update the list regularly (upon request when someone changed their repository.
- The authors present a guideline/checklist which could/should be implemented in future MIDL editions.
- The authors presented and discussed their results at the MIDL autumn academy and reached out to the community to receive feedback on their proposed checklist.
- The paper has a very nice discussion.


**Weaknesses:**

- Even though the paper gives a checklist for authors before submitting a paper, I am missing more solutions and references, e.g. further links to improve the reproducibility. If I am new in the field and what to submit a reproducible paper, how do I learn to do it?

**Deanonymize Review:**

no

**Detailed Comments:**

- The abstract is very detailed and is almost an introduction. I would recommend shortening it to focus on the most important points.
- Even though the results of Melis et al. sound very interesting, I wouldn’t call a paper from 2018 recent. It gives the impression that the current NLP systems are still outperformed by a simple LSTM. Since 2018 a lot has happened in the field.
- On page 6, a reference is missing (“When training a model on privately acquired data, we warn against privacy violation
attacks (?) where attackers could get information about the training data through the
trained model weights.”)


**Paper Type:**

validation/application paper

**Questions To Address In The Rebuttal:**

I would highly recommend performing the same evaluation performed for 2018-2022 also for the MIDL 2023 papers. I understand that it was so far not possible due to the double-blind review process. Therefore, I would like to ask the PCs for an extension for the camera-ready-version of this paper to include that analysis! Please reach out to the PCs and ask for it!
Please address the points from the detailed comments section.

---

### Official Review · Reviewer_YNKS · 2023-02-06

**Confidence:** 5
**Preliminary Rating:** 5
**Recommendation:** Oral

**Summary:**

This paper discusses an important, relevant and hot topic in ML research: reproducibility.
The paper shows that MIDL submissions from 2018 to 2022 have improved as a result of guidelines and more awareness, but that we are not there yet. The paper gives several concrete suggestions how to improve (e.g. a new set of guidelines for the MIDL conferences, a checklist (appendix A), and e.g. extra fields in OpenReview). The paper also shows that adding a good repository to your work helps the community and leads to more citations for a paper.
I personally also strongly believe that we need to improve as a community in this area, so I applaud the authors for submitting this paper and the efforts they have done.

**Strengths:**

- Relevant topic, thorough analysis, strong discussion section
- Transparancy in the results, making all the scores public in a Google spreadsheet
- Conducted a discussion with the MIDL conference organizers

**Weaknesses:**

- I am missing one important detail for public repositories: a license. This is important for future use of published code in academic, but also commercial use. Please discuss this.
- The authors describe a poll that they did during a tutorial at the MIDL Autumn Academy. The poll ask people about their experiences with code repositories and filled out by 81 of the participants (from an average total of 90 participants). Could the results of this poll be covered in a graph or table?
- I am missing a good analysis between the differences of the proposed checklist and initiatives by other conferences such as MICCAI or NeurIPS.
- Regarding reproducibility, medical image analysis challenges and benchmarks are also very important. So, public challenges are also an important aspect in this discussion which I think deserves discussion in this paper.

**Deanonymize Review:**

no

**Detailed Comments:**

Many fields struggle to get enough reviewers for journal and/or conferences. If reviewers also need to do a technical review of code repositories, the work increases. How to handle this? Can we come up with ways how to automate this? I think an automatic test of a public repo could be worthwhile, which for example checks whether a repo is empty, whether there is a requirements.txt for a list of dependencies and versions, whether there is a proper license, etc


**Paper Type:**

validation/application paper

**Questions To Address In The Rebuttal:**

Please address my concerns addressed in the weaknesses section.
Specifically, I think this needs to be added:
- Please discuss licenses
- Please discuss the role of public challenges
- Please compare the checklist of other checklist of guidelines of other conferences

---

### Official Review · Reviewer_TPkU · 2023-02-07

**Confidence:** 4
**Preliminary Rating:** 4
**Recommendation:** Oral, Poster

**Summary:**

The authors proposed a set of guidelines for machine learning-related research for medical imaging applications. They evaluated the submissions to all accepted full paper submissions to MIDL between 2018 and 2022 using a combination of already existing guidelines from neurips and miccai, and formed a metric of reproducibility. They also propose the reason why MIDL platform may take a stand in the field of reproducibility to help researchers. Areas where guidelines are in place (links for code and data) show an improving trend year by year.


**Strengths:**

A 5-point evaluations from a comprehensive checklist was proposed.

Proposed guidelines would help both authors improve and validate their codes, which is also benefit to reviewers.

The evaluations of reproducibility help to recognize more relevant concerns and the guidelines can be adjusted accordingly to better fit MIDL, and to better reflect the current state of machine learning research.


**Weaknesses:**

The authors should also consider the value of reproducibility during review period, and may introduce a survey at the end of the review period like the neurips society does.

The authors may also like to consider the acceptance rate of all papers w/o codes, not only focus on the accepted ones.

Joelle Pineau, Philippe Vincent-Lamarre, Koustuv Sinha, Vincent Larivière, Alina Beygelzimer, Florence d'Alché-Buc, Emily Fox, and Hugo Larochelle. 2022. Improving reproducibility in machine learning research (a report from the NeurIPS 2019 reproducibility program). J. Mach. Learn. Res. 22, 1, Article 164 (January 2021), 20 pages.

The abstract is too long to read.

**Deanonymize Review:**

no

**Detailed Comments:**

The authors may adjust the length of the abstract.

**Paper Type:**

validation/application paper

**Questions To Address In The Rebuttal:**

The authors should also consider the value of reproducibility during review period, and may introduce a survey at the end of the review period like the neurips society does.

The authors may also like to consider the acceptance rate of all papers w/o codes, not only focus on the accepted ones.

---

### Meta-Review · Area_Chair_1tNZ · 2023-02-20

**Recommendation:** Accept (Oral)
**Confidence:** 4

**Metareview:**

The reviewers agree on the importance of the paper, and have made several suggestions to the authors, which the authors have either addressed or are planning to address (if the data becomes available). Given the overall positive evaluations and scores, and the engaging discussion I am happy to recommend acceptance for the paper.